# The Impact of Compounds Extracted from Wood on the Quality of Alcoholic Beverages

**DOI:** 10.3390/molecules28020620

**Published:** 2023-01-07

**Authors:** Tomasz Tarko, Filip Krankowski, Aleksandra Duda-Chodak

**Affiliations:** Department of Fermentation Technology and Microbiology, Faculty of Food Technology, University of Agriculture in Krakow, ul. Balicka 122, 30-149 Krakow, Poland

**Keywords:** wood components, maturation, aged alcoholic beverages, extraction

## Abstract

The production of some alcoholic beverages very often requires the use of wood from various tree species to improve the quality parameters (smell, taste, and color) of the drink. The review discusses the types of wood used in the production of wines, beers, and flavored vodkas. Changes occurring in wood during the process of toasting barrels or wood chips are described. The compounds derived from wood that shape the aroma, taste, and color of alcoholic beverages are presented. Depending on their origin, they were classified into compounds naturally occurring in wood and those formed as a result of the thermal treatment of wood. Next, the influence of the presence of wood on the quality of alcoholic beverages was described, with particular emphasis on wine, beer, whisky, and brandy. The final part of the article presents innovative techniques using wood to impart qualitative features to alcoholic beverages.

## 1. Introduction

The taste and aroma of most alcoholic beverages depend on many factors, including climatic and soil conditions, the type of yeast used for fermentation, the quality of the raw material subjected to fermentation, and beverage production technology. Building the sensory characteristics of alcoholic beverages is really about obtaining a variety of desired chemical compounds. These compounds come from various stages of liquor production, but it is important that they are obtained in the right proportions [1]. One of the stages responsible for the formation of flavor components is the maturation of alcoholic beverages in wooden barrels made of various types of wood. The most commonly used by coopers are French and American varieties of oak [2]. Nowadays, the main purpose of using wood in the production of alcoholic beverages is the extraction of compounds from wood structure, which then affect the original sensory characteristics of the beverage [3]. During the maturation of beverages in the presence of a selected species of wood, a number of chemical reactions occur between the ingredients of the beverages themselves and the compounds extracted from the wood, which affect the complexity of the aroma of alcoholic drinks.

## 2. Wood in the Production of Alcoholic Beverages

Alcoholic beverages have been produced by man since ancient times and have been known in all civilizations. Initially, people made only non-distilled beverages, mainly beer and wine. Amphorae and clay jars were most often used for fermentation, maturation, and storage, but due to their high weight and impermanence, they have been replaced by more technically advanced wooden barrels. The beginnings of barrel production date back over 2000 years. Originally, barrels were produced mainly in northern parts of Europe, where climatic conditions prevented the use of clay as a raw material for the production of jars and other vessels used initially in the production of alcoholic beverages [4]. In the early centuries of our era, wood was not treated by the producers of alcoholic beverages as a source of interesting and valuable chemical compounds that shaped the taste and aroma of the beverages. Barrels, due to their durability and relatively easy availability, were treated only as vessels for the production and transport of alcoholic beverages [5]. From the 5th century CE, the main raw material for the production of barrels was oak wood, due to its strength, flexibility, ease of processing, and relatively low gas and liquid permeability compared to other types of wood. The high availability of oak in the vicinity of breweries and wineries also contributed to its use by coopers [6].

### 2.1. Contemporary Use of Wood in the Production of Alcoholic Beverages

The original use of wooden barrels, i.e., the production, storage, and transport of beverages, has now been significantly limited. Nowadays, tanks and containers made of acid-resistant steel are used for these purposes, and wood is mainly used as a factor supporting the sensory qualities (aroma, taste, color) of alcoholic beverages [5]. The main purpose of using wood in the fermentation technology is to improve the quality of the beverages obtained by extracting the compounds contained in it. The use of wooden barrels also promotes the micro-oxidation reactions associated with the diffusion of oxygen through the wood pores. These reactions significantly affect the final sensory and aromatic properties of alcoholic beverages [6,7].

Currently, wooden barrels are used in the maturation stage of selected alcoholic beverages, which are characterized by very high quality and usually unique sensory features. This stage is mainly associated with grape wines and distilled beverages, such as whisky, brandy, or cognac. For selected alcoholic beverages, it is necessary to use only new barrels that ensure a high degree of extraction of aromatic compounds. Other beverages, such as cognacs or cherries, require the use of barrels in which wines were previously aged. However, the contact surface of the drink with the wood during aging in the barrel is not large, which significantly extends the duration of the process. This, in turn, is also associated with large losses of beverages, especially distillates, due to their evaporation through the pores of the barrel wood. Therefore, wood chips or wooden staves are increasingly used to mature alcoholic beverages [6,8]. The elimination of barrels in favor of wood chips is mainly related to economic aspects. The production price of chips is many times lower compared to the price of barrels, partly due to the possibility of using in the production of chips parts of wood of lower value that cannot be used in the production of barrels [7,8]. Commission Regulation (EC) No 606/2009 of 10 July 2009 states that only oak wood of the *Quercus* genus, whether or not toasted, may be used for the production and aging of wine. The most frequently chosen types of wood for the production of both barrels and wood chips are oak and chestnut, which are approved for use in wine production by the OIV (International Organization of Vine and Wine) [9]. The key species used in the production of barrels are *Quercus alba*, *Q. garryana*, *Q. macrocarpa*, and *Q. stellate*, mainly used in the USA, while in Europe the following species are used: *Quercus petraea*, *Q. robur*, *Q. cerris*, *Q. suber,* and *Q. lyrata*. Chips are also produced from other wood species, e.g., *Robinia pseudoacacia*, *Prunus avium* L., *Castanea sativa*, *Fraxinus excelsior* L., and *Fraxinus americana* L., but their use for the industrial production of certain beverages is not allowed [5,7,10,11].

### 2.2. Aromatic Compounds Naturally Present in Wood

*Quercus* oak wood, used for the production of barrels and wood chips, consists of 45% cellulose, 25% hemicellulose, and 20% lignin [12,13]. The remaining 10% of wood consists of a small amount of inorganic compounds and the most valuable—from the point of view of the production of alcoholic beverages—substances extracted from wood during maturation, which give the drinks a unique aroma, taste, and color [14]. These extractives include volatile and non-volatile steroids, esters, alcohols, terpenes, terpenoids, lactones, and phenols [12]. Wood, depending on the species and the type of treatment it was subjected to during the production of barrels or chips, is characterized by a wide spectrum of compounds that may affect the quality of beverages made with its use [15]. Wood is a rich source of phenolic aldehydes, volatile phenols, and phenolic ketones, and their concentration depends on the botanical origin, the degree of toasting (burning), and the part of the tree from which it comes [8]. These polyphenols are basically divided into three groups: phenolic acids, volatile phenols, and ellagitannins. Due to their origin, phenolic acids are divided into derivatives of benzoic acid (such as gallic, syringic, salicylic, and vanillic acids) and cinnamic acid (like caffeic, ferulic, and sinapic acids) [13]. From a botanical point of view, the polyphenols found in oak wood perform physiological functions, e.g., they participate in protective mechanisms against ultraviolet radiation or harmful pathogens. However, in the context of alcoholic beverage production, they play a superior role in creating their desired sensory characteristics [13].

Low-molecular-weight phenols are responsible, among others, for imparting color, bitterness, and astringency to beverages, they also increase the antioxidant potential of beverages [16]. 4-Vinylphenol was found in oak and gives alcoholic beverages an aroma described as “stable” and “medicinal” [13,17]. In turn, 4-ethylphenol, another phenol derivative found in oak, had horsey, leathery, and sweaty saddle-like aromas [18]. Guaiacol and 4-methylguaiacol, which have smoky and spicy aromas, have olfactory thresholds of 25 and 65 μg/L, respectively. 4-Ethylguaicol, with a 33 μg/L olfactory threshold, also contributes spicy, toasted, and smoky aromas. Vinylguaiacol, with an olfactory detection threshold of 40 μg/L, contributes spicy, clove-like, and oak aromas [13,19].

Ellagitannins are a group of compounds that have protective functions in wood. Their content in the dry matter of oak can reach 10%. They give the wood the appropriate hardness and limit the action of harmful microorganisms [16]. In addition, among phenolic compounds, they are the most easily extracted by alcoholic beverages during their maturation. Castalagin and vescalagin are ellagitannins that are most abundant in oak wood, and their content can vary from 40 to 60% of the weight of all ellagitannins [20]. There are also numerous lyxose/xylose derivatives, such as grandinin and roburin E, as well as dimeric forms, i.e., roburin A, B, C, and D [21] (Figure 1). Ellagitannins belong to the compounds of oak wood and, due to their antioxidant activity, affect the quality parameters of beverages [22]. Their amount in wood depends mainly on geographical origin, age, species, and sampling site [13,23]. The species of oak that are most often used in the production of barrels include *Quercus alba*, *Quercus robur* and *Quercus petraea*, however, it is the French variety (*Quercus robur*) that has the highest content of ellagitannins [22,23]. The ellagitannins contained in the wood are responsible for the astringency and bitterness of the beverages made with it. In the case of astringency, they show a low threshold of sensory perception by humans, at the level of 0.2–6.3 μmol/L, which is related to their spatial structure [21]. The astringency sensation caused by ellagitannins differs from that caused by flavanols; it is described as milder, less expressive, and longer lasting in the mouth [22].

Carpena et al. [4] believe that from a sensory point of view, the most important compounds found in wood are lactones, mainly the *cis* and *trans* forms of β-methyl-γ-octolactone, which are responsible for the aromas of coconut and wood. Lactones have a significant sensory effect on alcohols matured with wood, e.g., brandy and whisky; hence, they are often called whisky lactones [4]. The research of Ghadiriasli et al. [12] shows that oak wood is also a source of other lactones, including γ-octolactones, γ- and δ-nonalactones, and γ-decalactones with a peach scent.

Table 1 summarizes the compounds derived from wood that have the greatest impact on the sensory properties of alcoholic beverages.

As mentioned earlier, oak wood is mainly used in the production of alcoholic beverages, but wood species other than oak are also used, e.g., chestnut (Castanea sativa), acacia (Robinia pseudoacacia), or cherry (Prunus avium) [5]. The species of wood used as well as its origin largely affect the composition of the wood and thus, the amount and profile of the compounds that shape the aroma of alcoholic beverages. Table 2 presents the occurrence of selected compounds in wood, depending on the species.

The table shows the composition of wood that has been seasoned outdoors for 24 months. Such wood is a good material for the production of later roasted barrels and various forms of wood chips roasted to varying degrees [5,38]. Literature data show a large variation in the content of compounds included in wood, which is mainly related to its origin. Different oak species are usually richer in volatile phenolic compounds compared to other wood species (Table 2). Among this group, the chestnut tree stands out because it is rich in eugenol and isoeugenol. Lactones are only found in oak wood. Furanic compounds, phenolic aldehydes, phenolic ketones, and tannins and ellagitannins dominate in oak and chestnut wood. Cherry wood contains much fewer compounds that affect the aroma of alcoholic beverages only phenolic acids are present in a relatively high concentration (Table 2). In acacia wood, very few compounds valuable from the point of view of the maturation of alcoholic beverages are observed [5,7,37].

### 2.3. Aromatic Compounds Formed as a Result of Thermal Treatment of Wood

Toasting the wood is one of the wood preparation processes during the production of oak barrels. This procedure has an extremely intensive effect on the qualitative and quantitative changes of the compounds that can be extracted during the maturation of alcoholic beverages [39]. Toasting wood increases the amount of compounds that contribute to the smoky flavor of beverages. This is due to the thermal degradation of the compounds contained in the wood. The conditions of the toasting process also have a significant impact on the decomposition of wood components or the formation of new aromatic compounds [4].

Depending on the temperature used and the time of its impact, the degree of toasting can be divided into three levels: slight (light) toasting, medium toasting, and heavy (high) toasting [5]. Light toasting is achieved by thermal treatment of the wood at approx. 120 °C for up to 90 min. Under these conditions, only lignins and hemicellulose are gradually degraded, while cellulose is not affected at all. In the case of medium toasting, the temperature does not exceed 230 °C, while high toasting is obtained using a temperature above 230 °C for more than 15 min [21]. Toasting the wood affects only the top layer of the wood, which is a maximum of 2 mm. This is due to the poor thermal conductivity of wood, which means that only the surface layers of wood undergo thermal changes. The deeper layers of wood remain unchanged, and therefore their chemical composition depends only on the natural characteristics of the wood. Thermal treatment of wood leads to a series of chemical reactions, as a result of which numerous volatile and non-volatile compounds are formed that affect the final aroma of the wood itself and, consequently, the organoleptic character of beverages matured with the use of wood [13].

Pyrolysis and hydrothermolysis are to some extent responsible for the degradation of wood components, including lignins, hemicellulose, and ellagitannins, which are easily hydrolyzed [40] (Figure 2). As a result of the thermal decomposition of lignins (Figure 3) and hemicelluloses (Figure 4), volatile phenols, phenolic aldehydes, ketones, and acids are formed, which, subjected to further dehydration, are converted into lactones. Phenolic compounds typically impart smoky, woody, and vanilla flavors to beverages. On the other hand, polysaccharides produce furan compounds, giving the aroma of roasted almonds, nuts, caramel, or bread [4,5]. Chira et al. [20] reported that the intensity of wood toasting may affect the content of ellagitannins and their structure. Depending on the temperature used, these compounds will be more or less degraded. In addition, at a toasting temperature of 165 °C, the castalagin derivative, dehydrocastalagin, reaches its maximum content in oak wood, while exceeding the temperature of 190 °C causes a significant decrease. The most abundant volatile compounds in toasted oak wood are: 2-furaldehyde, 5-methylfurfural, whisky lactones (in *cis* and *trans* forms), with smaller amounts of guaiacol, *p*-cresol, eugenol, *cis*-isoeugenol, and vanillin [23,41,42].

Chatonnet et al. [39] showed that toasting has a significant effect on ellagitannins naturally present in oak wood. This treatment carried out at high temperatures often results in a decrease in the content of these compounds, although the light and medium degrees of toasting the wood do not significantly affect their content. In addition, the researchers indicate that the samples subjected to heavy toasting did not contain roburin D and E. High toasting temperatures also increase the content of ellagic acid as a result of the thermal degradation of ellagitannins. In addition, the high processing temperature significantly reduces the antioxidant activity and the ability to scavenge free radicals in the tested samples [20,23,40,41]. In the case of volatile compounds, the opposite tendency was observed than in the case of ellagitannins [40]. This may be due to the increase in the content of eugenol and vanillin as a result of the thermohydrolysis of lignins and the generation of their precursors and the oxidation of fatty acids, which are responsible for the increase in the content of whisky lactones. In turn, Cadahía et al. [42] found that wood toasting had the greatest effect on phenolic compounds. The use of a medium degree of toasting (30 min, 160–170 °C) resulted in an increase in the content of phenolic aldehydes and acids, while the presence of guaiacol, dimethoxyphenol, and cresol, responsible for the aroma of burnt wood, was absent. The increase in the content of phenols in wood is explained by the formation of cinnamic aldehydes and benzoic acids as a result of lignin degradation at temperatures from 120–165 °C. At higher temperatures (165–195 °C) these compounds are thermolyzed to phenols. Alañón et al. [15] showed that thermal treatment negatively affects the content of terpenes and norisoprenoids contained in wood. In the chips examined by them, almost complete disappearance of the content of these compounds and only a small amount of trans-β-ionone and vomifoliol were noted. Toasting can therefore greatly reduce the presence of floral and fruit flavors by thermally degrading the compounds responsible for them. On the other hand, the amount of furfural in wood, which is a product of hemicellulose breakdown catalyzed by acids, increases with increasing process temperature.

Some research results indicate that wet toasting of wood may also affect the formation of aromatic compounds. Duval et al. [44] showed that when wood is not wet toasted, high temperature significantly increases the concentration of furfural, guaiacol, and vanillin, whereas the content of these compounds is lower in wood soaked water. Water can hinder or delay the process of thermal degradation of wood biopolymers and lipids. It was also noted [4] that if the process was carried out at a temperature above 200 °C, water stimulated the formation of furfural and prevented the thermal degradation of eugenol and both *cis* and *trans* forms of whisky lactones.

There is also a large variation in the content of compounds affecting the aroma of alcoholic beverages depending on the origin of the oak wood (Table 3) [2].

It was shown [2,15] that the concentration of furfural and its derivatives, guaiacol and 4-methylguaiacol, and vanillin increased during toasting. The amount of oak lactones drastically decreased on toasting, although those of cis-oak lactone still remained high for American and French oaks. These results are in contradiction with the results obtained by other authors [34,36], which showed an increase in the concentration of lactones after toasting. A decrease in eugenol concentration was also found in American and French oak samples, but an increase in Hungarian and Russian oaks. According to the authors [2], this could have been caused by the overheating of the samples during toasting. However, the concentration of isoeugenol increased in all samples.

## 3. The Influence of Wood on the Quality of Alcoholic Beverages

### 3.1. Wine

The consumer value of wine consists of many factors, primarily sensory parameters, i.e., aroma, taste, and color. The quality of wines is largely shaped by oenological processes performed both during fruit cultivation and wine production [5,45]. The composition of fruit, in particular the content of sugars, organic acids, polyphenols, including anthocyanins, and many other ingredients that are precursors to flavor and aroma components, determines the quality and sensory characteristics of wines. The bouquet of wine is shaped mainly by volatile compounds formed during the fermentation and maturation of beverages [46,47,48]. The presence of wood during the maturation of wines can also largely shape their quality parameters [5,49,50,51]. Most of the wines highly rated by consumers, especially red wines, owe some of their specific sensory characteristics to aging in contact with wood [52]. This stage is often a key factor in the creation of high-quality wines. It is during this process, thanks to the wood used, that a number of physical and chemical changes occur that affect the aroma compounds naturally present in young wine, but also that many components contained in the wood are extracted [53,54]. In the aging process, there are many variables that affect the quality of the wine obtained, including aging time and the quality of wood used for the production of barrels or chips. The following factors also have a significant impact: the content of ethyl alcohol in the wine, the method of aging (static or dynamic), and the ratio of the contact surface of the barrel to the volume of the wine [8,51]. In turn, the degree of wood toasting will affect the profile and concentration of compounds that shape the aroma of wines. Untoasted or lightly toasted wood will release more hydrolyzed ellagitannins, which may also help stabilize the red color of the wine. Medium and high degrees of toasting will stimulate the passage of smoky and spicy phenols into the wine [5,53]. The wine maturation in a wooden barrel is associated with the occurrence of oxidation processes, which result in a change in color, loss of the character of the used fruit variety, and the development of new aromas [48]. This is due to micro-oxygenation, i.e., passing a small amount of oxygen through the barrel. This leads to the condensation of anthocyanins and tannins contained in the wine but also to the release of phenolic compounds from the surface of the wood [48,55]. These reactions directly affect the color and astringency of the wine and modify its aroma. In particular, ellagitannins accelerate the process of condensation between anthocyanins and tannins, e.g., vescalagin and malvidin, which results in a change in wine color from violet-red shades, characteristic of young red wines, to brown-red shades, typical of mature wines [8,56]. In addition, ellagitannins react with the products of condensation of flavanols and anthocyanins, and this modulates the organoleptic properties of wine, in particular astringency, because these compounds interact with salivary proteins [57,58].

In the case of red wines, compounds derived from wood, such as furfural, guaiacol, whisky lactones, eugenol, vanillin, and syringinaldehyde, have the greatest impact on their aroma parameters. Furfural, as one of the most important compounds of woody origin, is responsible for giving wine the flavor of dried fruit and roasted almonds [59]. Bautista-Ortn et al. [60] showed that the use of oak wood (barrels and chips) during maturation primarily increases the content of furfural in wine. This tendency was maintained until the 6th month of aging, after which the content of this compound decreased with the extension of the maturation time, which may be related to the biological conversion of furan aldehydes to the corresponding alcohols. Similar trends were observed in the case of 5-methylfurfural however, its concentration decreased after 3 months of aging. It was also shown that the use of oak barrels increased the concentration of 5-methylfurfural in wines compared to wood chips [60].

Whisky lactones derived from oak wood occur in two stereoisomers, *cis* and *trans*, which give the wine different aromatic notes. The *cis* form is responsible for weak coconut, earthy, or sometimes moldy notes, while the *trans* form gives the wine spicy, celery, or coconut notes. In winemaking, whisky lactones are usually associated with coconut, vanilla, and spicy aromas, but they may change depending on the grape variety used [61]. Guaiacol and its derivatives are mainly formed as a product of lignin decomposition during the wood toasting process and are responsible for the roasted aromas of wine [59].

The concentration of guaiacol and 4-vinylguaiacol depends mainly on the time of maturation, as well as on the type and dose of oak chips. These compounds can also be used as an indicator of the degree of wood toasting due to their chemical origin. In addition, 4-vinylguaiacol and 4-vinylphenol can be formed in the process of enzymatic or thermal decarboxylation from cinnamic acids [54]. Esters are considered to be extremely important compounds in shaping the sensory properties of wines; they usually contribute to the formation of fresh fruit and floral notes. Fermented fruit contains a small amount of esters; their presence in wine is related to enzymatic and chemical reactions that occur during the fermentation and aging of wines. Some of the esters identified in wines also come from precursors extracted from wood during aging, and their concentration in wine can vary greatly. For example, the content of ethyl furoate, ethyl cinnamate, and ethyl vanillate significantly depends on the degree of toasting of the wood and increases with the lengthening of the maturation time. These esters shape the heavy coconut and cinnamon-vanilla aromas of the wines. On the other hand, esters responsible for light floral aromas, such as 2-phenylethyl acetate and ethyl decanoate, are usually decomposed or interesterified during the aging of wines, and their concentration is reduced [62].

The approval of chestnut wood for winemaking has resulted in an increase in the variety of aromas in wines. The use of chestnut in the aging of wines may promote the formation of more phenols and gallic acid compared to oak wood. Chestnut wood is also characterized by greater oxygen permeability than oak wood, which results in a low content of catechins, epicatechins, and anthocyanins in the wine aged with its use, which most likely undergo accelerated polymerization. Chestnut gives the wine fruity notes, and in the case of oak wood, the aromas of vanilla, smoky, and spicy are more noticeable [52].

The typical color of red wines is due to the presence of many anthocyanins and their derivatives. Unfortunately, the total anthocyanin content decreases diametrically with wood aging. The reason for this gradual decrease in monomeric pigments is the reaction of the native grape anthocyanins with the constituents of the wood and with the oxygen passing through the micropores of the wood. These changes are manifested by a decrease in the intensity of the color of the wine and differences in shade compared to wines not aged in the presence of wood [63].

White wines were traditionally rarely aged in barrels or with oak chips. It was believed that this procedure lowered their quality due to the masking of the original fruit and floral notes found in white wines. Esters, alcohols, and terpenes are responsible for these specific floral and fresh notes of white wines [64,65]. Therefore, white wines need a special aging procedure to avoid the roasted and bitter aromas of furan compounds derived from toasted wood [66]. Moreover, in the context of white wines, the extraction of compounds contained in wood also favors negative processes responsible for oxidation, which consequently leads to a change in their color. The darker, brown color of wine is not a desirable aspect because it is often perceived as a technological defect in the eyes of consumers [67]. Nowadays, the use of wood for the production and maturation of white wines is a common procedure that also enables the creation of alcoholized wines with a full flavor and complex aroma [64,68]. Nikolantonaki et al. [69] showed that the aging of white wine in the presence of oak wood contributed to a significant increase in the concentration of extractive substances from wood: ellagitannins almost four times, phenolic acids almost two times, and flavan-3-ols more than four times. These changes depended on the aging time and the degree of toasting of the wood but did not adversely affect the quality of the wines. Although one of the most suitable white grape varieties to ferment the must and age the wine in oak barrels is Chardonnay [70], there are many publications describing the aging of other white wines, such as Verdejo [71], white Listán [72], Muscatel [66], Sauvignon blanc [73], Encruzado [74], or Malvazija istarska [75]. It has also been shown that the use of wood species other than oak (chestnut, acacia, and cherry) for the maturation of white wines significantly reduces the level of furfural and 5-methylfurfural, as well as guaiacol and its derivatives [51].

### 3.2. Beer

The production of beer became a part of the culture and tradition in many countries (e.g., Czech Republic, USA, Germany, and Great Britain). Before the advent of stainless steel, barrels were a typical vessel used for the storage and transportation of beer. Some styles of beer, e.g., porter, often had to be aged for about 6–12 months in wooden tanks in order to soften their harsh taste [76].

Some types of beer are closely associated with aging in barrels. The Belgians used, and continue to use, the process of enriching the flavor composition with barrels. Currently, around the world, there is a so-called beer revolution, where beer production in small, craft breweries is rapidly developing. The increased development of this branch of industry has forced producers to compete more, which is why breweries try to distinguish themselves primarily by the originality of their products [77]. For this reason, as in the case of wines, there is a growing interest in using wood to produce characteristic beverages with interesting aromatic qualities that can be obtained during their maturation in barrels or in the presence of chips. Beer maturation can be carried out either in new barrels or in barrels that were previously used for the production of wine, whisky, or other beverages. In the latter case, the beer can acquire the taste and aroma of the beverage that was previously in the barrel [78]. This gives some types of beer a richer, more appealing taste. In addition, IPA beers have traditionally been stored and transported in barrels [78].

Today, in search of new, interesting, and attractive flavor compositions for beers, these beverages are barreled or other methods are used to obtain a flavor composition derived from wood. In the case of beer, malt and hops are the ingredients responsible for providing the aromatic compounds associated with the character of a particular style of beer [79,80]. Contact with wood can enrich beer with flavor notes associated with vanilla, sharpness, coconut, a smoke component, or the wood itself. A significant increase in the concentration of gallic acid, 5-HMF, and furfural was found in beers aged in oak wood cubes compared to beers aged in bottles. A strong degree of toasting especially increased the concentration of gallic acid and 5-HMF, while a weak level of toasting increased the concentration of furfural almost 10-fold [78]. The method of introducing aromas associated with wood should be balanced so that it does not dominate over the proper characteristics of the beer [78,81]. Maturation of beers in the presence of oak cubes worsened their sensory evaluation [78]. Other studies [82] showed that aging beers in the presence of wood, depending on the type of wood, increased the content of polyphenols and the aroma of burned, condimented, vanilla, tannic, and resinous flavors. Most wood did not interfere with the malty flavor. The exceptions are oak and amburana woods, but both woods were rated as most aromatic by the panelists, suggesting that the malty aroma was hidden amongst the others. Phenolic compounds are usually responsible for the aroma of wood. During the maturation of beers with wood, the concentration of many monophenols increases, and this increase is more pronounced when more wood chips are added [81,83]. Moreover, as pointed out by Sterckx et al. [83] levels of compounds such as 4-vinylguaiacol and 4-ethylguaiacol were found to be unrelated to the species or amount of wood used during aging. The extraction of wood-derived monophenols was favored rather by the low pH of the beer and the high alcohol content. In the case of craft beers using oak chips, an increase in the total phenol content was observed, which also resulted in higher antioxidant activity of these beverages. However, in the case of commercial beers, such increases were not observed, which is due to the use of filtration using PVPP (polyvinyl polypyrrolidone), which binds low-molecular phenolic compounds responsible for causing browning and bitterness. It should be taken into account that extractable compounds, such as polyphenols, may negatively affect the colloidal stability of beer, causing haze as a result of the formation of protein-polyphenol complexes.

In addition, the use of wood in the production of beer may be associated with the loss of some compounds as a result of their absorption by wood or as a result of evaporation [84]. Quite popular in the tradition of Belgian brewing is the use of wooden barrels as a source of various microorganisms, including *Brettanomyces* yeast and *Lactobacillus* and *Pediococcus* bacteria. They reside mainly in the pores of barrel wood, where they are difficult to remove during washing, but their presence is necessary during the production of traditional sour beers. This is also the reason for maintaining appropriate sterility in the production of commercial beers. In relation to other styles, these organisms are responsible for beer defects, which makes wood chips an interesting alternative to barrels. The use of chips allows for faster maturation effects associated with taste sensations that would not be possible to obtain in barrels [76].

### 3.3. Whisky

Whisky is produced mainly from barley in a process that includes stages such as mashing, fermentation, distillation, and maturation. It is thanks to this last stage that the whisky acquires its final character, which is appreciated all over the world. It is from the wooden barrel that the compounds responsible for the aroma and specific color pass into the drink. Aging in oak barrels also increases the antioxidant potential of whisky by increasing the concentration of phenols, furans, and acidic compounds [85,86,87]. Storing fresh distillates in oak barrels for a long period of time, even several dozen years, causes their natural sharp and raw aroma to be changed into mild, pleasant notes, and the clear distillate takes on an amber color, which suggests a number of changes in the whisky and the production of flavor ingredients and pigments [88]. Maturation is a key factor that gives distillates the right quality and organoleptic profile by stimulating physicochemical interactions between the wood and the liquid, which leads to a change in color and volume as well as the content of ethyl alcohol in the drink. However, the final taste of the beverage will depend mainly on the initial composition of the distillate [89].

The method and time of maturation vary greatly and are often determined by local or international laws depending on the final product. Another factor influencing this process is the ratio of wood surface to volume of liquid, so the use of smaller barrels can have a significant impact on increasing the extractivity of compounds contained in wood [90]. Compounds present in alcohol distillates may react with compounds present in wood during maturation, which is favored, among others things, by the high content of ethyl alcohol contained in them. The main aromatic changes during maturation are the result of oxidation, reduction, polymerization, polycondensation, and esterification reactions. The concentration of fatty acid ethyl esters increases, while in the case of 3-methylbutylacetate, its concentration decreases due to transesterification [90]. Other compounds responsible for the aroma of whisky are lactones, phenols, nitrogenous bases, sulfur compounds, and carbonyl compounds [91]. A characteristic of oak-matured drinks is the presence of eugenol, which provides a clove-like flavour, and whisky lactone, which has been found to be directly correlated to assessments of whisky quality. Studies have reported a direct relationship between assessments of whisky quality and the level of 5-hydroxymethylfurfural, phenolic compounds like vanillin, and possibly hydroxymethyl-pyranones [92]. Aging of whisky in oak barrels also increases its antioxidant activity due to the presence of numerous phenolic acids, which is affected by the conditions of the process and the quality parameters of the barrels [91].

### 3.4. Brandy

The types of brandy include flavored vodkas made from many different raw materials, primarily from grapes, but also from other fruits. What raw material will be used to produce brandy depends mainly on the region, its availability, and its price. For this reason, there are significant differences in quality, taste, and smell between particular types of brandy [93]. Brandy is defined as a spirit drink produced by the distillation of wines with an ethanol content below 94.8% vol., the aging process of which includes the use of oak barrels for a minimum of one year. A particularly recognizable type of brandy is cognac, a traditional French alcohol produced exclusively in the Charante department. It is made as a result of the double distillation of white wine and its maturation for several years in oak barrels, thanks to which it gained the name eau-de-vie (French for water of life). Young cognac, as well as other brandy vodkas, have a high alcohol concentration of around 60–70% vol. before aging, are colorless and usually have floral and fruity aromas because they are rich in volatile compounds, mainly esters [24,94]. They acquire the right aroma, taste, and color during maturation as a result of a series of chemical reactions between the ingredients of the drink. Of key importance in the aging of brandy is the presence of wood, from which volatile and non-volatile wood compounds are extracted. These include lactones, furan compounds, vanillin derivatives, polyphenols, e.g., phenolic acids, coumarins, or lignins [24,93]. Phenolic acids are involved in the aging process of brandy and affect its sensory profile. For example, gallic acid, a product of the hydrolysis of soluble gallotannins and ellagitannins of oak wood, acts as a catalyst and removes sulfur particles. During maturation, the alcohol reacts with the surface of the oak barrel, which causes the cleavage of the aryl-alkyl ethers of lignins and the formation of vanillin, syringaldehyde, and their acids [95]. The woody notes of oak bring syringol, a volatile phenol formed as a result of thermal decomposition of lignins, and β-methyl-octolactone, but its two isomers present in brandy are responsible for other aromas [33]. It is interesting that when chestnut wood is used, these isomers do not penetrate into the drink, which is related to the composition of the wood. The *cis* isomer is responsible for sweetness and coconut notes, while the *trans* isomer is responsible for floral sensations. The highest intensities of caramel and burned notes were found in the brandies aged in chestnut wood. The toasting level of the barrels had a significant effect on the majority of the aroma descriptors, namely fruity, vanilla, woody, spicy, caramel, burned, smoke, green, tails, and glue. Brandies aged in heavily toasted barrels presented the highest intensities of vanilla, woody, spicy, caramel, burned, and smoke and the lowest intensities of fruity, green, tails, and glue [33]. According to Gadrat et al. [96] the degree to which the wooden barrel has been toasted also significantly influences the organoleptic differences of the cognac. The use of slightly toasted wood results in obtaining lighter and sweeter scent notes than in the case of heavily toasted wood, where smoke and leather aromas dominate [97].

A specific type of brandy is Brandy de Jerez. It is a spirit produced in Southern Spain and obtained from wine spirits and distillates that were aged in <1000 L oak barrels previously seasoned with Sherry wine. Brandy de Jerez, in addition to typical compounds extracted from wood, also contains other compounds that come from sherry wine, such as tartaric, lactic, or succinic acids [98]. In those cases, the barrels act as transfer vectors between the sherry wine that had been previously contained in the barrel and the newly aging distillate [99]. However, the typical compounds contained in oak wood are released into brandy in a much smaller amount than during their maturation in new barrels [100]. Hic et al. [3] showed that maceration of brandy with chestnut trees leads to the highest increase in antioxidant capacity compared to other types of wood (oak, acacia, apricot, mulberry, plum, cherry, and grape vine).

Micro-oxygenation accelerated the changes taking place in cider distillates when compared to traditional aging in barrels. Furthermore, a higher degree of brandy oxidation led to a higher concentration of benzoic derivatives, total acetaldehyde, oak lactones, and gallic acid and a more pronounced decrease in the levels of 3-methyl-1-butyl acetate and 2-phenylethyl acetate [101]. Caldeira et al. [102] showed that the best results for the sensory overall quality of the aged wine spirits were obtained when the oxygen flow was applied in the aging process over a longer period of time: 2 mL/L/month during the first 60 days, followed by a flow rate of 0.6 mL/L/month until the end of the experiment, a total of 365 days.

### 3.5. Extractivity of Aroma Compounds from Wood by Alcoholic Beverages

The qualitative changes of alcoholic beverages (wine, beer, distillates) described above that occur under the influence of contact with wood during maturation are influenced to some extent on the concentration of ethanol in these beverages. The drink penetrates the wood and washes out various compounds that shape the aroma, taste, and color of the drink. Variables here include the use of new or used barrels, toast levels, char levels, how the wood was seasoned, as well as the storage conditions during maturation. Water and alcohol penetrate the wood structure and initiate the hydrolysis of the hemicelluloses and lignin. The possibility of compound migration from wooden barrels or chips is related, among others, to the concentration of alcohol and the method of wood processing [43,103]. Table 4 presents the changes in the concentrations of compounds contained in various alcoholic beverages (beer, wine, and grape pomace distillate) without maturing and after aging, both without contact with wood and in the presence of toasted wood.

The aging of alcoholic beverages without the addition of wood, with some exceptions, had almost no effect on the level of volatile compounds (Table 4). The presence of wood significantly changed the profile of compounds responsible for the aroma and taste of beverages [103]. The increase in the concentration of furan compounds was statistically significant only for beer and wine, while the changes in the grape pomace distillate were masked by their initial high concentrations. The maturation of alcoholic beverages in contact with oak chips increases the concentration of syringaldehyde and sinapaldehyde [81,83]. The alcohol concentration has a significant effect on the extraction of lactones and vanillin (Table 4). In the case of distillates, the amount of lactones, especially the cis-isomer, was clearly higher than in beers and wines. Many papers highlight the effect of the concentration of ethanol in the aged beverage on these compounds [60,83,104].

Apart from extraction of volatile compounds, compound losses inherent to subtractive ageing were also observed as a direct result of wood application. Concentrations of ethyl acetate and ethyl decanoate decreased in all beverages after contact with wood. In the distillate, hexyl acetate, 2-phenylethyl acetate, and ethyl dodecanoate were adsorbed on the wood, mainly due to hydrophobic interactions [99].

## 4. Innovative Techniques Using Wood to Give Quality Features to Alcoholic Beverages

### 4.1. Ultrasound

Ultrasound is defined as sound waves with a frequency exceeding the limit of hearing for the human ear. In the food industry, waves with frequencies from 20 kHz to 10 MHz are used. The most important task of ultrasonic treatment is to modify the structure of the raw material, thanks to which the intensity of mass exchange is strengthened and chemical and biological changes occur that affect the properties of the final product [105]. Ageing on lees is a technique traditionally used in white and sparkling wines, but it may also be interesting for red wines. The autolysis of yeast leads to the release of cellular proteins, nucleic acids, lipids, and polysaccharides, and wines aged on lees are enriched in volatile aromatic compounds [106]. Wines aged with wood chips and treated with ultrasound (400 W, 24 kHz, 5–170 min per week) did not show statistically significant changes in total polyphenol content, color intensity, or volatile acidity, but significant changes in the content of volatile compounds were observed; an increase in the concentration of methanol, propan-1-ol, acetic acid, and most esters, and a decrease in the concentration of 2-methyl-1-butanol, 3-methyl-1-butanol, and 2-phenylethanol were found [107]. Tao et al. [108] showed that the use of ultrasound in the range of 6.3–25.8 W/L during the maturation of model wine accelerated the extraction of polyphenolic compounds. The authors also point out that high levels of ultrasound are supposed to promote the ultrasonic modification of wine composition and recommend the use of lower doses. Taloumi and Makris [109] noticed an acceleration of the extraction of components from chestnut wood from Tsipouro distillate which was ultrasonically-treated. The improved color and increased concentrations of phenolic compounds were also observed in ultrasound-treated plum distillates [110]. These changes depended on the type and dose of the wood chips as well as on the maturation conditions, e.g., heating at 35 °C. Ultrasonics (40 W/L, 40 kHz, 6 min/day) applied to the maturation of brandy with medium-toasted pieces of wood resulted in an increase in the extraction of phenolics by 33.9% after seven days of aging [111].

### 4.2. High Presure

In many cases, efforts are made to shorten the maturation time of alcoholic beverages and to reduce costs by replacing barrels with wood chips. A technique for reformulating alcoholic beverages can be the use of high pressure (HHP). Tao et al. [112] showed that wines with oak chips subjected to high pressure (250–650 MPa) were characterized by a higher concentration of polyphenols and anthocyanins and a stronger antioxidant potential than the control samples (without pressure). However, it was noticed that wines with high pressure were characterized by a weaker sweet and floral aroma and a stronger artificial taste. Other studies [113] report that slight changes occurred in the phenolic composition and color properties of red and white wines immediately after HHP treatments. In pressurized red wine, these changes manifested as a decrease in both total and individual phenolic compounds, while all color parameters increased. Additionally, the applied treatments resulted in the decrease of phenolic contents in white wine, with the exception of the increase of some free phenolic acids. However, the authors showed that wine treatment with HHP helped reduce the use of SO_2_ in wine production. Santos et al. [114] treated red wines with pressures of 500 MPa for 5 min and 600 MPa for 20 min. They showed the effect of HPP on the aroma and polyphenolic composition of wines, but it was observed only after 5 months of aging. A lower content of monomeric anthocyanins, phenolic acids, and flavonols in pressurized red wine after five months of storage was due to the increase in the condensation and oxidation reactions of these compounds; polymerization and cleavage reactions of proanthocyanidins also occurred. HHP can be potentially used as an enological practice to decrease the astringency of young red wines and increase some pleasant aromas [114]. The use of oak chips together with HHP shows the possibility of accelerating the aging process in certain types of wines like Cayetana white wine [115]. It allows obtaining in less than 10 min wines with similar physicochemical and sensory characteristics to those from a classical maceration in tanks, which needs at least 45 days.

### 4.3. Pulsed Electric Field

Pulsed electric field (PEF) is usually used for non-thermal inactivation of microorganisms [116]. In winemaking, PEF has been used to increase the extraction of anthocyanins from grape skins [117] and condensation between (+)-catechin and acetaldehyde [118]. The use of PEF during the maturation of alcoholic beverages with oak chips showed the effectiveness of this method. Agiorgitiko red wines treated with a pulsed electric field (1.2 kV/cm, 5 s) contained significantly more wood-derived compounds (vanillin, syringaldehyde, furfural, and oak lactone) relative to controls. In the case of whisky, the effect of PEF (5 kV/cm) increased the concentration of vanillin and syringaldehyde, and to a small extent, lactones, but in the case of brandy, PEF caused a decrease in the concentration of vanillin and furfural [119]. The use of a pulsating electric field of 1 kV/cm during the maturation of brandy in oak barrels enhanced the extraction of tannins, total phenols, and volatile phenols from the wood by about 50% [120]. The higher efficiency of extraction of compounds derived from wood in small barrels was also confirmed, which was associated with a larger contact surface of brandy with wood than in large barrels (225 L). The advantage of this technique is that it can be used directly in the barrel. The pulsed electric field has also been used to accelerate the formation of mannoproteins in wines [121]. Autolysis induced by PEF did not negatively affect the wines’ physicochemical properties. The mannoproteins released in a shorter time from PEF-treated cells had similar functional properties in wine as mannoproteins released during natural autolysis from untreated yeasts. As a result, this technique allows for the acceleration of “aging on lees”. The wine that received yeast treated with PEF released mannoproteins in one month, which was equivalent to wine with untreated yeasts in six months.

## Figures and Tables

**Figure 1 molecules-28-00620-f001:**
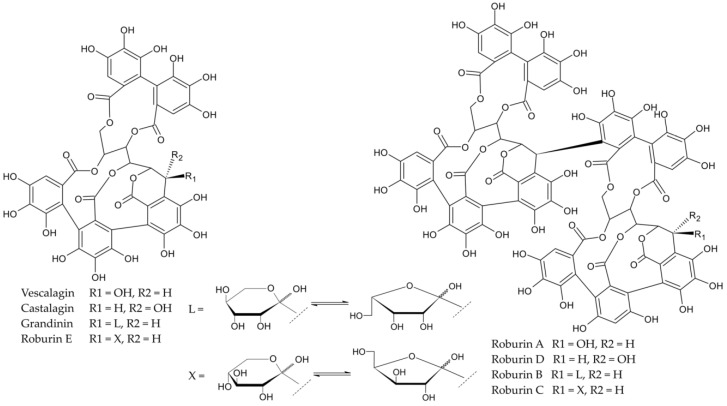
Chemical structures of the eight C-glucosylated ellagitannins. L—Lyxose, X—Xylose [24].

**Figure 2 molecules-28-00620-f002:**
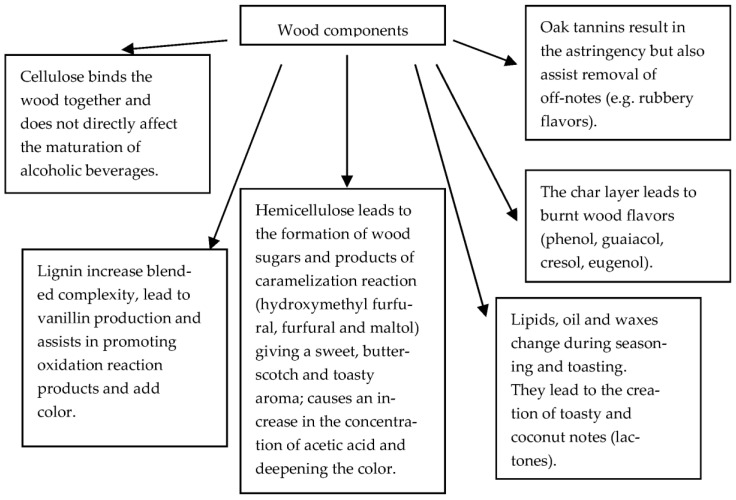
Major wood constituents and basic chemistry [43].

**Figure 3 molecules-28-00620-f003:**
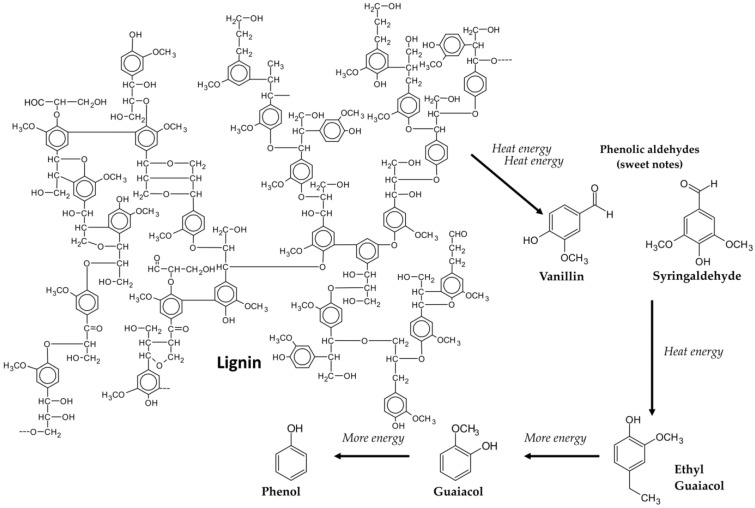
Thermal degradation of lignin [43].

**Figure 4 molecules-28-00620-f004:**
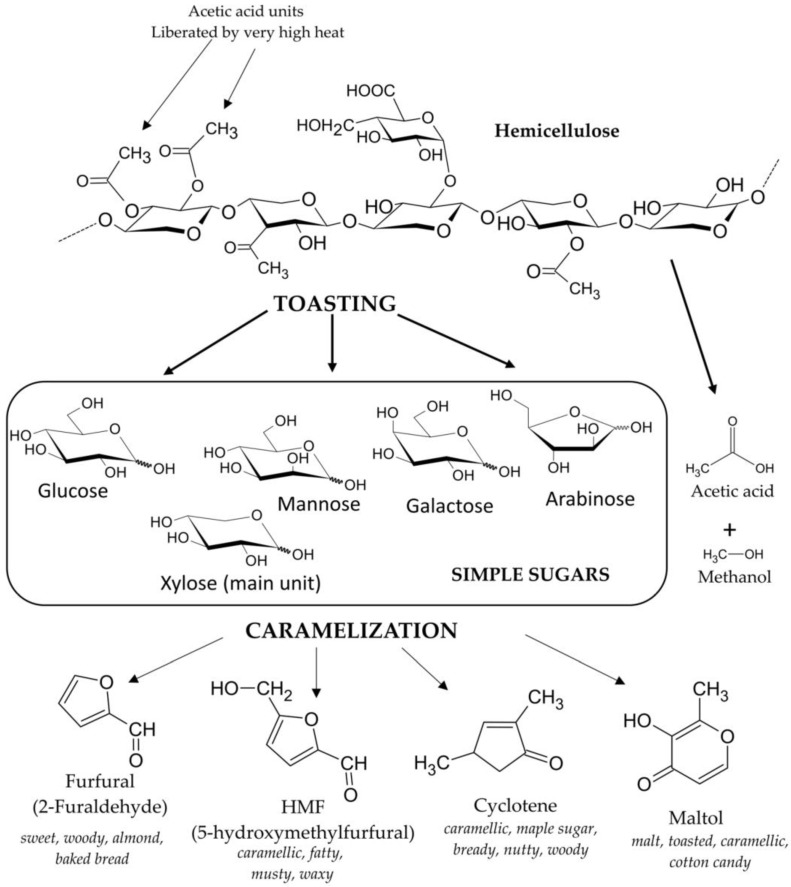
The structure of a hemicellulose molecule and the products liberated during toasting and caramelization processes [43].

**Table 1 molecules-28-00620-t001:** Main aroma compounds derived from wood [2,5,25,26,27,28,29,30,31,32,33].

Common Name	IUPAC Name	Aroma Notes	Olfactory Threshold
**Volatile phenols**			
Guaiacol	2-Methoxyphenol	smoke, sweet, medicine	9.5 µg/L
4-Ethylguaiacol	4-Ethyl-2-methoxyphenol	phenolic, smoked, leather	47 µg/L
4-Methylguaiacol	4-Methyl-2-methoxyphenol	spicy, phenolic, light green	20 µg/L
4-Vinylguaiacol	4-Vinyl-2-methoxyphenol	clove	40 µg/L
Eugenol	2-Methoxy-4-(prop-2-enyl) phenol	clove, honey, spicy, cinnamon	6 µg/L
Isoeugenol	1-Methoxy-4-(prop-2-enyl) phenol	floral, clove, woody	6 µg/L
Syringol	2,6-Dimethoxyphenol	smoke, burned, wood	570 µg/L
**Furanic compounds**			
Furfural	2-Furancarboxaldehyde	bread, almond, sweet	15 mg/L
5-Methylfurfural	5-Methyl-2-furancarboxaldehyde	almond, caramel, burnt, sugar	16 mg/L
Maltol	3-Hydroxy-2-methyl-4H-pyran-4-one	honey, toasty, caramel	5 mg/L
5-Hydroxy-methylfurfural	5-Hydroxymethyl-2-furaldehyde	caramel	100 mg/L
**Lactones**			
*trans*-β-Methyl-γ- octalactone	*trans*-4-Methyl-5-butyldihydro-2-(3H)-furanone	coconut, woody, vanilla	140–370 µg/L
*cis*-β-Methyl-γ- octalactone	*cis*-4-Methyl-5-butyldihydro-2-(3H)-furanone	coconut, woody, vanilla	20–46 µg/L
**Phenolic aldehydes/Phenyl ketones**			
Vanillin	4-Hydroxy-3- methoxybenzaldehyde	vanilla	1 mg/L
Syringaldehyde	4-Hydroxy-3,5- dimethoxybenzaldehyde	vanilla	50 mg/L
Acetovanillone	1-(4-hydroxy-3- methoxyphenyl)ethanone	vanilla	1 mg/L

**Table 2 molecules-28-00620-t002:** Compounds [µg/g] found in different wood species, seasoning for 24 months (open air) [5,7,34,35,36,37,38].

Compounds	*Quercus* *pyrenaica*	*Quercus alba*	*Quercus* *petraea*	*Quercus* *robur*	*Castanea* *sativa*	*Robinia* *pseudoacacia*	*Prunus avium*
Guaiacol	nd–0.5	0.0–3.3	nd–4.5	0.1	0.1–0.2	0.1–0.9	0.2–0.5
4-Ethylguaiacol	0.0–0.1	nd	nd–0.0	0.1	0.0	0.1	0.0
4-Methylguaiacol	nd–0.4	0.1–1.5	nd–0.6	0.2–0.7	0.1–0.2	0.0–0.1	0.0–0.1
4-Vinylguaiacol	0.9–2.2	0.2	0.2–1.0	1.0–1.3	0.3	0.4	0.3
Eugenol	nd–7.3	1.4–6.0	1.1–6.5	1.1–1.6	2.0–4.5	0.1–0.9	0.1–0.1
Isoeugenol	nd–1.4	1.1–1.8	0.7–4.3	0.3	2.2–2.4	1.0–3.4	0.0–0.6
Syringol	1.1–1.7	0.1	0.1–0.2	0.2	0.1–0.3	0.8–1.9	0.4–1.5
Furfural	3.2–11	1.2–5.8	3.4–12.9	8.9–11.0	5.5–6.7	0.5–0.9	nd–0.7
5-Methylfurfural	0.2–1.0	0.2	0.1–0.2	1.1–2.0	0.1–0.2	0.0–0.1	0.0–0.1
Maltol	0.3–1.4	0.5	0.2–0.8	0.3–0.4	1.1–2.0	0.9–1.4	0.4–0.5
5-Hydroxy-methylfurfural	0.9–13.0	0.4–6.3	0.3–4.0	0.8–2.7	14.0–21.0	0.2–0.5	0.2–0.5
*trans*-β-Methyl-γ- octalactone	5.3–34.0	1.6–5.0	2.1–15.0	3.4–4.0	nd	nd	nd
*cis*-β-Methyl-γ- octalactone	5.3–68.0	22.0–37.4	6.1–56.0	2.8–23.0	nd	nd	nd
Vanillin	1.6–32.6	6.8–309.8	2.0–45.7	9.3–94.8	17.0–71.8	1.7–4.7	1.1–37.8
Syringaldehyde	4.2–104.7	16.0–52.8	2.7–514.7	14.0–218.	38.0–240.1	6.0–10	2.6–52.8
Acetovanillone	0.6–0.7	0.4	0.4	0.7–1.0	0.4–0.5	0.2–0.3	0.2–0.3
Butyrovanillone	0.1–3.1	1.5	1.4–1.9	1.7–2.5	1.9–2.0	0.7–1.0	0.4–1.0
Gallic acid	43.0–96.2	44.1–105.3	36.0–108.0	36.3–374.3	189.3–349.3	ns	2.6–116.1
Ellagic acid	66.5–219.0	132.1–277.0	109.4–198.3	84.4–227.8	74.0–140.4	ns	10.5–22.1
Protocatechuic acid	59.2–74.3	34.5–63.6	79.9–269.5	118.1–435.7	88.5–154.4	ns	8.1–114.6
Vanillinic acid	6.0–31.0	30.5–61.2	67.5–142.9	0.0–17.1	40.0–66.7	ns	25.0–34.7
p-Coumaric acid	26.5–51.9	14.5–48.7	13.4–43.3	2.7–135.7	48.3–154.2	ns	4.5–28.0
Sinapic acid	58.7–81.7	32.4–38.5	143.3–728.0	102.9–185.7	207.3–225.3	ns	47.2–146.1
Syringic acid	26.7–48.9	30.2–53.6	83.9–349.1	47.0–120.6	57.5–84.6	ns	37.5–48.3
Caffeic acid	1.2–2.8	3.5–5.6	4.9–15.0	3.5–5.0	3.4–5.1	ns	6.1–15.6
Ferulic acid	3.4–8.4	4.1–15.8	9.6–22.3	3.0–21.0	10.1–16.0	ns	11.7–18.5
Roburin A	47.8–68.5	27.8–38.1	42.8–53.5	72.7–108.8	40.4–52.0	ns	0.0
Roburin B	32.3–50.7	17.8–27.8	29.8–40.6	77.4–101.7	16.2–25.4	ns	0.0
Roburin C	34.5–56.1	19.3–30.1	30.9–45.8	127.8–206.1	11.1–18.5	ns	0.0
Grandinin	137.9–200.4	65.4–81.0	87.5–134.5	278.9–457.1	26.9–43.4	ns	0.0
Roburin D	64.1–88.7	21.7–34.9	42.3–66.7	150.3–222.0	22.5–31.0	ns	0.0
Vescalagin	486.0–728.5	124.8–166.3	315.4–484.2	739.4–1096.7	1725.4–1998.7	ns	0.1–19.0
Roburin E	380.6–604.2	117.3–173.2	326.6–493.6	652.1–942.5	843.4–1089.9	ns	0.0–8.8
Castalagin	1054.2–1641.9	321.4–467.2	757.6–1067.8	1042.0–1660.7	1485.3–1932.1	ns	8.8–127.7

nd—not detected; ns—not studied.

**Table 3 molecules-28-00620-t003:** Concentration (μg/g) of some oak wood-derived volatile compounds in toasted and untoasted American, French, Russian, and Hungarian oak species [2,15].

Compounds	American		French		Hungarian		Russian	
	Untoasted	Toasted	Untoasted	Toasted	Untoasted	Toasted	Untoasted	Toasted
Furfural	23.8	139.8	18.7	186.4	21.9	234.2	20.8	114.2
5-Methylfurfural	0.96	16.1	0.81	50.1	1.45	29.9	1.43	30.3
2-Methylphenol	0.02	0.09	0.03	0.19	0.03	0.24	0.02	0.12
4-Methylphenol	0.08	0.20	0.01	0.24	0.03	0.46	0.03	0.22
Guaiacol	1.02	4.88	0.21	5.82	0.16	12.3	0.07	3.81
Nonanal	2.11	1.08	1.30	0.41	1.70	0.67	1.08	0.58
*trans*-2-Nonenal	0.80	0.49	1.86	0.86	4.12	1.51	2.46	0.98
4-Methylguaiacol	0.05	4.71	0.04	5.35	0.04	11.7	0.03	4.93
Decanal	0.83	0.46	0.55	0.25	0.93	0.42	0.45	0.36
*trans*-β-Methyl-γ- octalactone	30.4	9.67	14.0	5.39	0.08	0.02	0.06	0.02
*cis*-β-Methyl-γ- octalactone	104.1	39.3	96.3	32.4	0.29	0.09	0.37	0.08
Eugenol	7.71	3.82	4.52	3.50	2.41	5.33	0.6	2.57
Vanillin	0.62	3.77	0.07	1.28	0.16	3.81	0.03	2.96
*cis*-Isoeugenol	Tr	0.76	Tr	0.58	Tr	0.94	Tr	0.76
4-Propylguaiacol	0.01	0.38	0.01	0.16	0.01	0.57	Tr	0.29
*trans*-Isoeugenol	0.09	4.81	0.25	2.74	0.08	5.80	0.03	4.01
Syringaldehyde	0.06	0.10	0.03	0.11	0.04	0.77	0.02	0.24
Maltol	0.05	0.86	0.06	1.62	0.06	1.34	0.06	1.22

Tr—traces.

**Table 4 molecules-28-00620-t004:** Concentration [µg/L] of selected compounds in beer (B), wine (W), and grape marc spirit (S) in their initial states, after 48 h at 40 °C without wood (B40, W40, S40), and after 48 h at 40 °C with reused wood (BW40, WW40, SW40) [103].

Compound	B	B40	BW40	W	W40	WW40	S	S40	SW40
ethyl butyrate	41.0	40.7	46.8	106.6	100.0	111.2	1605.2	1677.0	1566.0
isoamyl acetate	988.0	982.8	1029.7	580.1	604.3	556.3	4703.7	5001.5	4690.9
ethyl hexanoate	137.9	140.7	126.6	257.5	288.3	235.6	6393.0	6867.9	5964.1
ethyl lactate	nd	nd	59.0	3454.7	3576.2	3907.3	1297.2	1411.6	1660.5
ethyl octanoate	129.0	133.4	68.0	237.5	316.3	152.8	24,531.9	26,831.5	20,902.4
ethyl decanoate	18.6	20.7	11.1	30.0	33.7	11.7	25,557.4	32,014.4	11,350.1
diethyl succinate	nd	nd	666.1	7562.9	7296.9	8087.6	8252.1	8317.3	8551.1
2-phenylethyl acetate	788.9	809.4	735.0	128.4	119.6	106.1	904.2	913.4	752.3
diethyl malate	nd	nd	35.3	151.4	150.2	200.6	nd	nd	54.8
2-methyl-1-propanol	140.7	139.5	198.5	749.1	869.3	821.5	5820.2	5929.5	5984.5
2-methyl-1-butanol + 3-methyl-1-butanol	5781.5	5787.7	7796.4	20,560.5	23,359.9	23,400.2	103,415.4	116,358.1	128,263.2
1-octanol	6.9	7.4	8.6	18.4	17.2	16.9	1481.4	1478.1	1414.6
benzyl alcohol	11.1	13.1	17.8	154.0	104.2	124.5	228.1	231.1	281.1
2-phenylethanol	6027.0	6477.9	7893.9	9337.1	9286.7	9969.5	9463.2	10,462.5	11,003.2
linalool	8.9	9.9	9.5	11.7	10.4	8.8	391.6	391.63	368.3
α-terpineol	5.6	7.4	7.6	34.9	33.1	30.0	861.6	889.9	864.0
β-citronellol	nd	nd	nd	nd	nd	nd	558.7	540.2	503.0
cis-nerolidol	nd	nd	nd	nd	nd	nd	377.3	355.9	315.3
acetic acid	40.0	83.4	46.6	303.4	298.6	305.0	nd	nd	nd
isovaleric acid	17.8	20.6	21.5	91.9	93.9	92.8	1272.2	1345.5	1505.3
hexanoic acid	295.1	214.4	327.0	742.8	660.2	742.6	3721.9	1833.9	1976.4
octanoic acid	1896.3	2040.3	2182.2	1606.8	1555.1	1574.4	13,013.4	14,085.0	12,475.1
decanoic acid	444.8	462.8	257.7	466.8	460.1	292.2	17,977.0	16,888.1	17,033.0
furfural	84.7	102.9	342.3	76.0	96.8	320.6	4207.7	4518.0	4894.0
5-methylfurfural	nd	1.5	109.7	nd	nd	133.6	678.8	719.4	783.5
5-hydroxymethylfurfural	13.1	16.7	24.6	nd	nd	nd	nd	nd	nd
cis-oak lactone	nd	nd	364.2	nd	nd	359.4	nd	nd	548.6
trans-oak lactone	nd	nd	66.3	nd	nd	144.1	nd	nd	163.3
γ-nonalactone	38.3	42.3	39.2	11.5	10.6	10.6	233.8	231.6	210.3
benzaldehyde	nd	5.1	16.3	nd	nd	nd	2337.5	2427.9	2295.7
vanillin	15.6	14.7	459.0	nd	nd	480.8	nd	nd	542.1
syringaldehyde	nd	nd	2513.2	147.3	317.0	2573.9	nd	nd	2812.3
sinapaldehyde	nd	nd	2377.4	nd	nd	2258.2	nd	nd	4012.7
4-methylguaiacol	nd	nd	18.7	nd	nd	nd	nd	nd	68.6
4-ethylguaiacol	0.8	0.6	4.4	nd	nd	nd	229.2	263.2	263.4
eugenol	nd	nd	17.6	1.3	5.9	14.1	456.2	423.6	422.1
4-ethylphenol	nd	nd	nd	nd	nd	nd	281.8	236.4	249.4
4-vinylguaiacol	94.8	104.3	38.2	nd	nd	nd	nd	nd	nd
2,6-dimethoxyphenol	nd	nd	71.4	31.5	32.5	112.3	nd	nd	nd
4-vinylphenol	11.2	16.2	5.5	nd	nd	nd	nd	nd	nd
acetovanillone	8.3	11.3	90.9	nd	nd	nd	nd	nd	nd

nd—not detected.

## Data Availability

Not applicable.

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
