# Peer review of "The Impact of Compounds Extracted from Wood on the Quality of Alcoholic Beverages"

_molecules, 2023, doi:10.3390/molecules28020620_

Round 1

Reviewer 1 Report

The manuscript is very well written and scientifically appealing, as it provides an overview regarding some of the major physical-chemical changes occurring during wine, beer, whisky and brandy maturation in wood barrels. Still, it would be interesting to include one or two schemes or images highlighting some of the mechanistic changes discussed throughout the text in more detail.

Also, I just detected one minor error in line 170 (two "and"). One must be removed.

Reviewer 2 Report

I am carefully reviewing the manuscript “How the compounds extracted from wood impact on the quality of alcoholic beverages?”. In this manuscript, compounds derived from wood, which shape the aroma, taste and color of alcoholic beverages were reviewed. This topic is interesting and practical for the brewing industry. However, I do not think the authors well considered it and carefully written this review. From my opinion, this manuscript is not suitable to be published in the journal of molecules .

Some comments.

1. The title was How the compounds extracted from wood impact on the quality of alcoholic beverages?. However, the article only presented the aroma compounds with and without thermal treatment, no non-volatile compounds derived from woods such ellagitannins, terpenes and polysaccharides were mentioned. Furthermore, only compounds were listed. It did not explain how it impact.

2. In part two, the mechanisms how wood products impact wine supposed to be presented. But, what is the difference of extracted compounds among the woods species? How thermal treatment impact the flavor compounds? How the seasoning treatment impact the compounds extracted from wood? These questions were not well presented.

3. In part three, more techniques concerned to accelerating aging process. No involvement with woods products. E.g. Reference 102, 106, 107, 108, 112.

4. For a good review, well prepared tables and figures are really needed.

Reviewer 3 Report

This manuscript is an interesting review on the effect that using wood (either barrels or chips) may have on the taste and other organoleptic properties of alcoholic beverages. It is wirttrn in a very clear way, easy to follow and gives a comprehensive information on the topic. My felicitations to the authors.

The are only a few minor comments that the authors may have in consideration.

First, it is important to state the novelty of the work, and how this review adds new or different information to the available literature.

I suggest to change the title of the paper. In its present form, the reader could expect an analysis of chemical mechanisms by which the different compounds extracted from either barrels or chips alter, modify or improve the quality or the sensorial perception of the beverages. But actually the paper does not come into those details, but lists the compounds responsible for those changes. "The impact of compounds extracted from wood on the quality of alcoholic beverages" is perhaps more descriptive.

The paper is a long text with only a table, which renders the manuscript in some way hard to be followed; my suggestion is to include at least another table where the main wood derived compounds are listed as a function of either the alcoholic beverage, the effect on it (e.g., lower astringence or others), adding aromas and so on; or another criterion.

Abstract. Line 10. Add "among others", because in line 16 authors mention whisky and brandy (but not vodka)

- A couple of typing mistakes detected:

Title of section 2. Correct "alcohoic"

Line 126. Correct "dute"

Round 2

Reviewer 2 Report

accept